# Sustainability of the Agricultural Systems of Indigenous People in Hidalgo, Mexico

Doris Leyva [1], Mayra De la Torre [2] and Yaxk'in Coronado [1,*]

[1] Catedra-Conacyt-Unidad Regional Hidalgo, Centro de Investigacion en Alimentacion y Desarrollo A. C. Ciudad del Conocimiento y la Cultura de Hidalgo, Blvd. Santa Catarina S/N, San Agustin Tlaxiaca 42163, HG, Mexico; doris.leyva@ciad.mx

[2] Unidad Regional Hidalgo, Centro de Investigacion en Alimentacion y Desarrollo A. C. Ciudad del Conocimiento y la Cultura de Hidalgo, Blvd. Santa Catarina S/N, San Agustin Tlaxiaca 42163, HG, Mexico; mdelatorre@ciad.mx

* Correspondence: yaxkin.coronado@ciad.mx

**Abstract:** Agricultural sustainability depends on complex relationships between environmental, economic, and social aspects, especially with small farm holders from indigenous communities. This work was centered on two municipalities of Hidalgo State in Mexico, Ixmiquilpan (mainly irrigated systems) and El Cardonal (rainfed systems). Our objective was to understand the relationships between the small farm holders and their agricultural systems. We evaluated the sustainability of their agricultural systems and made some recommendations. We applied the Framework for the Evaluation of Management Systems using Indicators (MESMIS, Spanish acronym); thirty-one indicators were identified, and quantitative indexes were established to assess the sustainability. The results showed that adaptability was a critical factor for irrigated and rainfed systems, and the main problem identified was youth migration. Additionally, the access to water and economic resources and the management of environmental resources are necessary in order to increase the yield of agricultural crops. Therefore, a holistic approach that considers the organization of small producers and synergy between indigenous knowledge and modern technologies is required for the territorial development of the communities.

**Keywords:** sustainability; MESMIS framework; rural agricultural systems; migrant remittances

## 1. Introduction

Sustainability science is a dynamic and growing field of research and practice [1]. In addition to academic terms, it also includes strategic efforts to connect science with practical decision making and implementation [2]. Sustainability is the attempt to balance the quality of human life while considering environmental, economic, and social objectives [3]. Since agriculture provides most of the world's food and fabrics, as well as papers and material for construction, the sustainable development of this sector is of utmost importance [4].

The FAO defines sustainable agricultural development as "the management and conservation of the natural resource base, and the orientation of technological and institutional change in such a manner as to ensure the attainment and continued satisfaction of human needs for present and future generations. Such development conserves land, water, plant and animal genetic resources, is environmentally non-degrading, technically appropriate, economically viable and socially acceptable" [5].

The expansion of conventional agricultural techniques, especially monoculture, and the massively increased use of agrochemicals have caused an environmental crisis at the global scale and have raised the necessity of new approaches in order to solve this crisis, such as sustainable agriculture. The main objectives of sustainable agriculture are: (i) improving the health of producers and consumers (food security, organic agriculture); (ii) maintaining the stability of the environment (biological methods of fertilization and pest

management); (iii) ensuring long-term benefits for farmers; (iv) considering the needs of current and future generations [6,7]. Traditional farming systems could be an option—for example, rural indigenous communities in Central Mexico combine polyculture (corn, beans, squash, chili, and other crops) with organic fertilizers, no-tillage farming, no agrochemicals, and rainfall [8]. A comparison of the sustainability indicators of this traditional system and other agricultural systems will allow us to identify advantages and bottlenecks. Furthermore, socioeconomic and environmental indicators could serve as tools in planning and decision-making processes at the community and regional levels. These indicators belong to six main categories: production, resilience, adaptability, organization, social equity, and self-sufficiency [9–12].

Concerns about agricultural sustainability go farther than environmental conditions and changes in internal trade; they also include the nature of the agricultural crisis that most countries are going through [13]. However, there are considerable discrepancies in the translation of the most appropriate philosophical and ideological aspects of sustainability, as well as in what the most appropriate methodologies are for evaluating it. In this way, the evaluation of sustainability is affected by problems inherent to the multidimensionality of the concept itself, which includes ecological, economic, social, cultural, and temporal dimensions [14]. Therefore, the evaluation requires a holistic and systemic approach, where multi-criteria analysis predominates [15]. Moreover, sustainable management of natural resources (SMNR) has mainly focused on "sustainable agriculture", but several authors have argued that SMNR should be understood in a broader sense, including activities such as forestry, livestock production, fisheries, mining, and ecotourism [16,17].

We believe that the most adequate technology for the evaluation of sustainability of complex agricultural systems is the framework for the evaluation of management systems using indicators (MESMIS, a Spanish acronym for "Marco para la Evaluación de Sistemas de Manejo de Recursos Naturales Incorporando Indicadores de Sustentabilidad"). This framework proposes six cyclic steps: (1) characterization of the systems; (2) identification of the critical points that are linked to the sustainability attributes—for example, productivity, stability, reliability, resilience, adaptability, equity, and self-reliance; (3) identification and selection of indicators; (4) measurement and monitoring; (5) analysis and integration of data; (6) conclusions and recommendations [11,18,19]. Therefore, we used MESMIS to evaluate and compare the sustainability of rainfed and irrigated agricultural systems of the Otomi communities in two municipalities of the Hidalgo State of Mexico. These communities stand out because they are located in a region with low agricultural productivity, with a semiarid climate and volcanic soil, hillocks, ravines, thorny scrubs, and few streams [20]. In the last five years, the average annual precipitation was 140 milliliters in El Cardonal and 142 milliliters in Ixmiquilpan [21,22]. In both municipalities, an ancestral polyculture system, referred to locally as "milpa" [23], is the basis of rainfed agriculture, and includes corn, beans, squash, chili, and many other crops. In irrigated lands, in contrast, monoculture systems are typical. This region was selected due to its long tradition of polyculture farming and the presence of both irrigation and rainfed farming.

Thus, there is a need to develop methods for evaluating the sustainability and performance of agricultural systems, as well as for guiding actions and policies for the sustainable management and preservation of natural resources. In the book *Sustainability of the Systems for Management of Natural Resources in Andean Countries*, seven cases from Bolivia, Colombia, Ecuador, and Venezuela were analyzed with the MESMIS framework. The aim was to obtain information in order to design alternatives for the sustainable management of natural resources with special emphasis on biosphere reserves. In the Boyacá department of Colombia, three typical agrosystems were selected and classified based on life quality, biodiversity, and family cohesion according to the biophysics, biological, technological, and socioeconomic components; seventeen indicators were used. The results suggest an interdependence among agricultural practices, biophysical conditions, and the socioeconomic situations of rural families [24].

In this work, we evaluated the sustainability and performance of agricultural systems of indigenous communities in Hidalgo State, Mexico and compared two typical agricultural systems: rainfed and irrigation farms. We integrated thirty-one indicators of economic, social, and environmental aspects to assess the extensive dry-land farming in order to identify the main variables that affect sustainability on the dry lands at the center of Mexico. Indeed, the results could be very useful for strategies and public policies, since the data available in our country are outdated. We suggest that our main findings could be useful for similar regions in other countries.

## 2. Materials and Methods

Mexico's indigenous population is one of the two largest in the Americas; more than one in ten Mexicans speaks an indigenous language, and 5.7 percent of them live in Hidalgo State [25]. Our study area included 18 indigenous (Otomi or Hñähñu, as they refer to themselves) villages located in the municipalities of Ixmiquilpan and El Cardonal in the Hidalgo State, Mexico (Figure 1). Ixmiquilpan is located at 20°29′03″ N, 99°13′08″ W with an altitude of 1680 m above sea level. This is an area dedicated to irrigated agriculture, the main crops being forage, vegetables, and grains [26]; the water source for irrigation is wastewater from Mexico City. El Cardonal is located between the latitudes 20°24′58″ N and 20°46′31″ N and longitudes 98°55′54″ W and 99°10′46″ W, with an altitude between 900 and 2900 m above sea level. This is an area dedicated to rainfed agriculture and grazing livestock (mainly sheep); the main crops are agave, corn, and fruit trees [27–29]. Inhabitants of both municipalities belong to the Otomi culture; their villages are communities of high or very high marginalization and medium social backwardness.

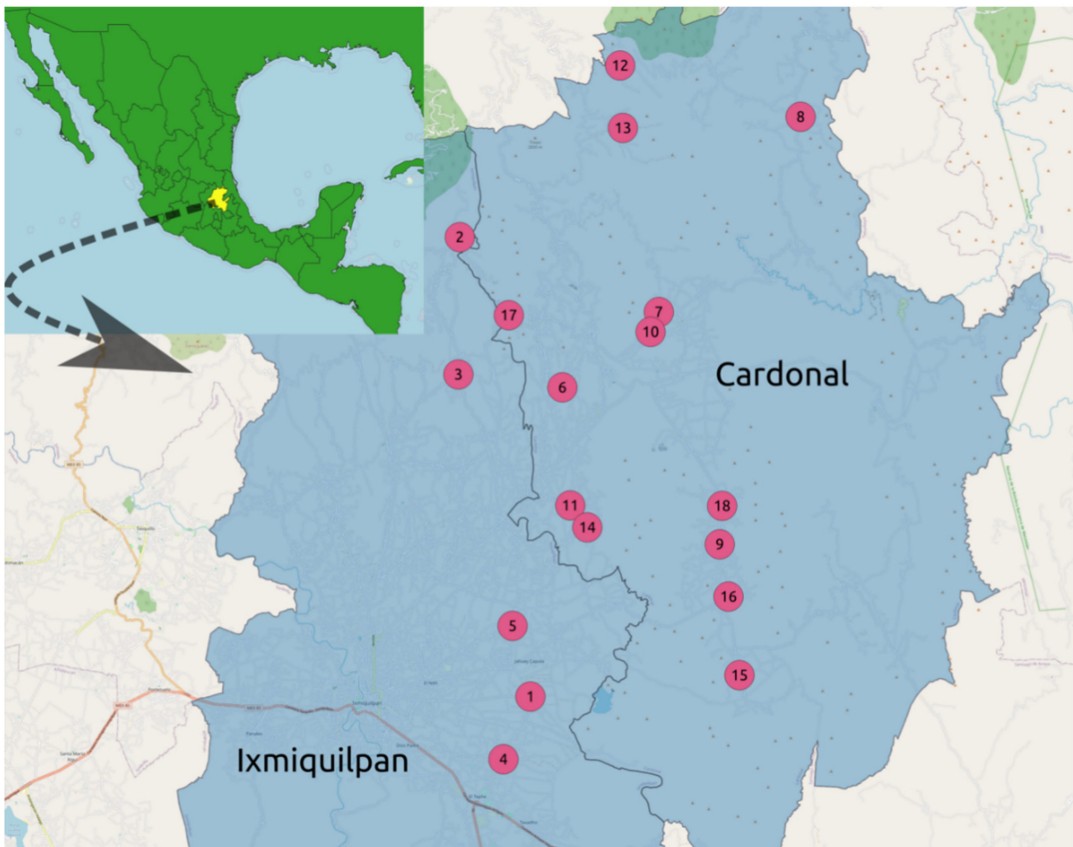

**Figure 1.** Location of municipalities and communities; Ixmiquilpan: (1) Bangandhó, (2) El Nogal, (3) El Olivo, (4) Pueblo Nuevo, and (5) San Pedro Capula. El Cardonal: (6) Cerro Colorado, (7) Chalmita, (8) Cieneguilla, (9) Durango Daboxtha, (10) El Bondho, (11) El Botho, (12) El Potrero, (13) El Tixqui, (14) Los Reyes, (15) Pozuelos, (16) San Andrés Daboxtha, (17) San Miguel Jigui, and (18) Santa Teresa Daboxtha.

We interviewed small farm holders who participated in the governmental "Territorial Development Program" (PRODETER, according to its acronym in Spanish), were supported by the Ministry of Agriculture and Rural Development (SADER, according to its acronym in Spanish), and lived in the municipalities of Ixmiquilpan or El Cardonal. We adapted a mobile application to study the social, environmental, and economic relationships of the agricultural systems. The mobile application koBoCollect v1.25.1 [30] allows the collection of quantitative and qualitative data through a series of questions, including multimedia files, such as photos, and GPS locations. Data collection was carried out during January and February 2020; using a non-probability test sample, we selected the producers in a random way while keeping a homogeneous geographical distribution so that each community was represented [31,32]. Between 15 and 20 percent of the producers registered in the database of PRODETER were interviewed (121), and all interviews were used. The data included socioeconomic characteristics, family structure, economic incomes and outcomes from agricultural and non-agricultural activities, land use, food crops and yields, use of crops (sale, self-supply), agricultural practices, farmers' awareness of climate change, use of chemical and biological pesticides, fertilizers, and herbicides, agricultural infrastructure, and adoption of new technologies. The product systems included in PRODETER were sheep, apple, and olive for Ixmiquilpan and corn, agave, sheep, and olive for El Cardonal. We selected farmers that had corn crops and included in our study all of the other agricultural products of the farms, including crops, livestock, fruit trees, etc. The Mexican government established a minimal number of farmers that had to participate in the PRODETER program in order to cover most of the indigenous farmers, who face problems of poverty and marginalization, as well as social backwardness. The government rules established that at least 15% of the producers had to be interviewed to have a statistically significant representation.

According to the MESMIS methodology, the first step was the characterization of the system by including socioeconomic and environmental features. The second step focused on the identification of the critical points related to the attributes of productivity, stability, reliability, resilience, adaptability, equity, and self-reliance. The third stage was the selection of sustainability indicators and their corresponding normalization, and they were flexible, easy to measure and to understand, and comprised social, economic, and environmental development (Table 1). The selection of indicators followed the structure proposed by the MESMIS methodology, and each indicator was calculated according to its specific class and normalized to one hundred. The fourth was measurement and monitoring, followed by analysis and integration of data. The last stage involved the conclusions and recommendations.

**Table 1.** Sustainability attributes, critical points, and indicators selected for the agricultural production systems.

| Sustainability Attribute | Critical Point | Indicator [1] | Reference |
|---|---|---|---|
| Productivity | Sales from agriculture producers | (a–b) Total agricultural sales | [24] |
| | Profitability | (c) Cost–Benefit ratio | [24,33] |
| | External supplies | (d) Agricultural supplies | [24] |
| Stability Resilience and Reliability | Water supply | (e) Water access | [24] |
| | Fertilizers | (f) Organic fertilizer | [10,11] |
| | Agrobiodiversity | (g) Diverse crops in the same agricultural field | [24] |
| | Type of seeds of corn | (h) Use of native and commercial seeds | [12] |
| | Other crops and fruit trees | (i) Native and commercial varieties | [12,24] |
| | Use of agroecological practices | (j) Agroecological practices | [24] |
| | Biocontrol | (k) Pests and diseases | [24] |
| | Sources of income | (l) Income diversification | [11,24] |
| | Extension services | (m) Access to extension services | [11,24] |
| | Farmers desertion | (n) Permanence of workers | [24] |

**Table 1.** *Cont.*

| Sustainability Attribute | Critical Point | Indicator [1] | Reference |
|---|---|---|---|
| Adaptability | Education | (o) Educational level | [24] |
| | Global climate change | (p) Aware of the global climate change | [24,33,34] |
| | Availability and reuse of soil and water sources | (q) Management of natural resources | [24] |
| | Generation renewal | (r) Transmission of traditional agricultural techniques | [33] |
| | Technological adoption | (s) Scientific and technological innovation | [24] |
| | Infrastructure | (t) Machinery availability | [33] |
| | Capacitation | (u) Agricultural advisory services | [24] |
| Equity | Agricultural workforce | (v) Permanent workforce | [24] |
| | | (w) Temporal workforce | [24] |
| | Infrastructure distribution | (x) Equipment and machinery distribution in the community | [33] |
| | Income distribution | (y) Gini coefficient | [33] |
| | Gender equity | (z) Women participation | [24] |
| | Food security level | (aa) Fractional food security level for maize | [33] |
| Self-management | Agriculture system inputs | (ab) Pesticides | [24] |
| | | (ac) Chemical fertilizers | [24] |
| | Agriculture system incomes | (ad) Average agriculture income | [24] |
| | Dependence of external inputs | (ae) Dependence of agrochemical and agriculture machinery | [33] |
| | Organization and participation | (af) Organization and participation | [33] |

[1] The indicators are the percentages of peasants for each parameter, with the exceptions of fractional food security for maize, permanent workforce, demand for temporary workforce, and income distribution.

For each indicator, we defined a parameter, classification, or calculation to quantify its representative value for sustainability. Each indicator was defined as follows:

(a) Total agricultural sales for irrigation systems were calculated by considering all agricultural sales. The parameters used for classification were farmers with lower sales than the second quartile of the farmers' total sales distribution, with sales between the third and second quartiles, and sales higher than the third quartile.

(b) Total agricultural sales for rainfed systems were calculated as described in (a).

(c) The Cost–benefit ratio (CBR) was calculated by considering all agricultural inputs divided by the total sales associated with agricultural activities for each producer. The parameters for classification were low for farmers with a CBR of less than one and high for those with a CBR of greater or equal to one.

(d) Agricultural supplies were the average of consumables and capital inputs. Three levels were defined: lower than the first quartile, between the second and third quartiles, and greater than the third quartile.

(e) Water access included the natural water sources and irrigation channels available for agricultural activities. The parameters were defined as minimum when only rainfall was available, average when two different sources of water were accessible, e.g., rainfall and irrigation channels, and high when three or more sources of water (rainfall, irrigation channels, and wells) were available.

(f) Three parameters were defined for fertilizers: use only of organic fertilizers, employment of both organic and chemical fertilizers, and no fertilizers.

(g) Agrodiversity was classified as low when it was monoculture farming, medium when two crops were intercropped, and high when more than two crops were intercropped.

(h) For corn seeds the parameters were defined as only native seeds, commercial seeds, and a mix of native and commercial seeds.

(i) For crops and fruit trees, native or commercial varieties were considered, and the definition was the same as that of (h). Backyard cultures were also included.

(j) For agroecological practices, the parameters were low when no agroecological practices were used, medium when only one practice was implemented (zero-tillage or biofertilizers), and high when two or more practices were adopted (zero-tillage, conservation tillage, biofertilizers).

(k) For pests and diseases, the parameter was defined as high when agricultural pests and diseases were present in corn and the intercropped crops, medium when only corn was affected, and low when they were not detected.

(l) Income diversification was calculated as the number of income-generating activities. It was defined as low when the income was only from agricultural activities, medium when the farmer carried out other remunerated activities besides agriculture (e.g., bricklayer, small merchant, etc.), and high when he or she had at least two more productive activities in addition to agriculture.

(m) Extension services involved giving smallholders knowledge of agronomic techniques and skills to improve their productivity, food security, and livelihoods. For access to extension services, the parameter was defined as high when farmers received extension services when they asked for them or even if they did not, since the Mexican government provided it, low when they did not receive them even when they asked for governmental extension services, and null when they did not receive assistance and they did not know with certainty if they needed it.

(n) For generational renewal, we considered three groups of agricultural workers: the first group included workers from 18 to 34 years old, the second group included workers from 35 to 60 years old, and the third group included workers older than 60 years old.

(o) Related to the educational level, farm holders were classified into three groups according to their educational levels: uneducated—the ones that did not attend school; the second group included those who had elementary or junior high school studies, and the third group included the ones with high school or college studies.

(p) For the impact of global climate change, we considered three groups of producers: those who were aware of global climate change and thought it had a big impact on their crops, a second group of farmers who thought global climate change did not affect their crops, and a third group of producers that did not know whether climate change affected their crops.

(q) The management of natural resources indicated The percentage of producers who practiced water and soil conservation. The parameter was classified as optimal if they practiced water and soil conservation, as medium if they practiced water or soil conservation, or as non-conservation.

(r) For generational transmission of knowledge, the age of farmers who implemented the traditional agriculture practices was taken into account. The parameters were high when the producers who implemented these traditional practices were from 18 to 34 years old, low when farmers from 35 to 59 years old implemented them, and null when farmers older than 60 years old used them; we assumed that the use of these practices was due to the transmission of knowledge to younger farmers.

(s) Technological adoption was considered as the percentage of producers who had access to cell phones, internet, etc. The parameters were high when the farmer used two or more instruments, e.g., cell phone and internet, for agricultural activities, low the when producer had access to only one instrument, and null when the producer did not use an instrument.

(t) Infrastructure was the percentage of farmers that had access to tractors and other farming equipment.

(u) Agricultural advisory service was defined as the percentage of farmers that received training.

(v) Permanent workforce was the number of family members who were permanent agriculture workers per hectare per year. The optimal indicator was fitted to three workers per hectare.

(w)　Demand for temporary workforce was the number of hired agriculture workers per hectare per year. The optimal indicator was fitted to three workers per hectare.

(x)　Distribution of machinery and equipment was the percentage of producers who owned tractors and/or farming equipment.

(y)　The Gini coefficient was used to assess inequality. It was defined as previously described [35,36]:

$$I_G = \frac{\sum_{i=1}^{N-1} P_i - q_i}{\sum_{i=1}^{N-1} P_i} \tag{1}$$

where $q_i$ is the cumulative relative income of the sample for each system and $P_i$ is the cumulative relative frequency of the population. In our case, $q_i$ was a farmer's cumulative relative income from agricultural sales and $P_i$ was the cumulative population, i.e., the number of members of the farmer's family. The Gini coefficient was calculated for the small farm holders that owned either rainfed or irrigated farmlands.

(z)　The participation of women was the percentage of female farmers with respect to the total number of farmers.

(aa)　Chemical pesticides corresponded to the percentage of farmers who used chemical pesticides.

(bb)　Chemical fertilizers corresponded to the percentage of peasants who used chemical fertilizers.

(cc)　The Fractional Food Security Level (FFSL) for corn was calculated as [9]:

$$FFSL = \frac{Y_C A_C}{200N}, \tag{2}$$

where $Y_C$ is the corn yield in Kg/ha, $A_C$ is the net corn area cultivated, and $N$ is the number of the farmers' family members. The level of corn food security in Mexico was 200 Kg per person per year [37].

(dd)　The total income was the average household income per farmer per year. The parameter was segmented into three levels: high for income over the third quartile, medium for income between the third and second quartiles, and low for income under the second quartile.

(ee)　External inputs were defined as the percentage of farmers that purchased agrochemicals and/or used farming machinery. The parameter was classified into three levels: high for two or more external inputs, medium for one external input, and low for farmers without external inputs.

(ff)　Organization was the percentage of farmers who were members of a farmers' association.

### 3. Results

*3.1. Characterization of the System*

The El Cardonal and Ixmiquilpan municipalities are mostly "Hñähñu" (Otomi) and speak Spanish and Hñähñu; they live in nuclear families, and only very few have extended families. The average age of the farmers was 51 years, and most of them had elementary studies; in contrast, their offspring and relatives had an average age of 30 years and had junior and senior high school studies [38]. Typically, the farmers' farmland extensions were less than one hectare in Ixmiquilpan and from 2 to 4 hectares in El Cardonal. Some farmers owned and cultivated up to three different agricultural lands. In El Cardonal, the agricultural lands were rainfed, while in Ixmiquilpan, 95% of the lands had irrigation systems. These irrigation systems used residual water from Mexico City. It is worth mentioning that in Ixmiquilpan, most of the agricultural lands were communal; in contrast, in El Cardonal, 70% were private properties. Even when the parcels were very small, they were not rented to other producers. Most notably, migration played an essential role in both municipalities; inhabitants migrated to other Mexican states and mainly to the USA in order to improve their living standards. In El Cardonal, 62% of the families had at least one

member who migrated, and in Ixmiquilpan 44% of the families had at least one member who migrated (Table 2).

**Table 2.** Socioeconomic indicators of the El Cardonal and Ixmiquilpan municipalities.

| Indicator | Level | Ixmiquilpan | Cardonal |
|---|---|---|---|
| Average age (years) | —— | 49 ± 11 | 53 ± 12 |
| Education (%) | College | 11.1 | 7.4 |
| | Senior high school | 27.7 | 7.4 |
| | Junior high school | 44.4 | 44.4 |
| | Elementary school | 16.6 | 25.9 |
| | Uneducated | 0 | 14.8 |
| Number of agricultural lands owned per farmer (%) | 1 | 22.2 | 25.9 |
| | 2 | 33.3 | 40.7 |
| | 3 | 22.2 | 25.9 |
| | 4 | 5.5 | 3.7 |
| | 5 | 5.5 | 3.7 |
| | 6 | 5.5 | 0 |
| | 8 | 5.5 | 0 |
| Agricultural land area (%) | 0–1 Ha. | 55.5 | 25.9 |
| | 1–2 Ha. | 22.2 | 0 |
| | 2–3 Ha. | 0 | 25.9 |
| | 3–4 Ha. | 5.5 | 25.9 |
| | 4–5 Ha. | 11.1 | 7.4 |
| | 5–8 Ha. | 5.5 | 14.8 |
| Agricultural land tenure (%) | Communal | 73 | 30 |
| | Private property | 27 | 70 |
| Estate in land (%) | Ownership | 94 | 93.4 |
| | Borrow | 2 | 3.3 |
| | Rent | 4 | 3.3 |
| Family members in household (%) | 2 | 16.6 | 33.3 |
| | 3 | 22.2 | 33.3 |
| | 4 | 50 | 25.9 |
| | 5 | 11.1 | 7.4 |
| Family that has at least one member who migrated (%) | Yes | 44.4 | 62.96 |
| | No | 55.5 | 37.03 |

*3.2. Sustainability Attributes and Critical Points*

3.2.1. Productivity

- Productivity: The total agricultural sales in the high class were higher for rainfed lands than for irrigated ones, although the corn yield was lower. Indeed, rainfed agricultural products included several crops, fruits, agave, livestock, and edible wild plants, as well as edible insects and backyard products. Notably, the inputs—including investment and agricultural supplies—for the irrigated and rainfed systems were similar (Table 3).

**Table 3.** Productivity indicators for the El Cardonal and Ixmiquilpan municipalities.

| Indicator | Class | Ixmiquilpan | Cardonal | [2] Sustainability Dimension |
|---|---|---|---|---|
| Total agricultural sales (irrigation) | [1] High (irrigation) >USD 4.7 K/year | 16 | 20 | E |
| | Medium (irrigation) USD 3.3 K/year–USD 4.7 K/year | 33 | 20 | E |
| | Low (irrigation) <USD 3.3 K/year | 50 | 60 | E |

**Table 3.** *Cont.*

| Indicator | Class | Ixmiquilpan | Cardonal | [2] Sustainability Dimension |
|---|---|---|---|---|
| Total agricultural sales (rainfed) | [1] High (rainfed) >USD 4.7 K/year | 33.3 | 22.7 | E |
| | [1] Medium (rainfed) USD 3.3 K/year–USD 5.9 K/year | 33.3 | 31.8 | E |
| | Low (rainfed) <USD 3.3 K/year | 33.3 | 45.4 | E |
| Cost–benefit ratio (CBR) | Low <1 | 27.7 | 29.6 | E |
| | [1] High ≥1 | 72.2 | 70.4 | E |
| Agricultural supplies | High >USD 1.7 K | 27 | 33.3 | E |
| | Medium USD 1.7 K–USD 1.3 K | 66.6 | 29.62 | E |
| | [1] Low <USD 1.3 K | 33.3 | 37.0 | E |

[1] Selected as an indicator of sustainability in Figure 2. [2] E: Economic dimension.

### 3.2.2. Stability, Resilience, and Reliability

- Water shortages, dry lands, and droughts obligate farmers to develop strategies and actions for keeping their crops, such as polyculture systems and the use of natural fertilizers. Usually, native corn was grown in rainfed systems and backyards. In addition, in irrigated farmlands, hybrid and commercial seeds were used. It is worth mentioning that the farmers with rainfed systems used agroecological practices, including minimum tillage, natural fertilizers, corn, intercropped crops, and pest-repelling plants (Table 4). Pests and phytopathogens are an important problem, especially in irrigated farmlands, but the identification and biological control of pests and diseases in both agricultural systems could increase resilience.

**Table 4.** Stability, resilience, and reliability indicators for the El Cardonal and Ixmiquilpan municipalities.

| Indicator | Class | Ixmiquilpan | Cardonal | [2] Sustainability Dimension |
|---|---|---|---|---|
| Water resources | [1] High (residual water and underground water) | 11.1 | 3.7 | EN |
| | Medium (residual water) | 72.2 | 14.81 | EN |
| | Low (rainfall) | 16.6 | 84.48 | EN |
| Fertilizers | [1] Organic | 100 | 92.6 | EN |
| | Mixed | 0 | 1.1 | EN |
| | Null | 0 | 6.3 | EN |
| Agrobiodiversity | Low (monoculture) | 61.1 | 11.15 | EN |
| | Medium (two crops) | 16.6 | 37.03 | EN |
| | [1] High (three or more crops) | 22.2 | 51.81 | EN |
| Corn seed variety | [1] Native | 73.3 | 87.5 | EN |
| | Hybrid | 26.6 | 12.5 | EN |

**Table 4.** *Cont.*

| Indicator | Class | Ixmiquilpan | Cardonal | [2] Sustainability Dimension |
|---|---|---|---|---|
| Variety of other crops and fruit trees | [1] Native | 29.7 | 59 | EN |
| | Hybrid | 62.16 | 34 | EN |
| | Mixed | 8.1 | 6.8 | EN |
| Agroecological practices | Null | 16.6 | 29.62 | EN |
| | Medium | 27.7 | 25.92 | EN |
| | [1] High | 55.5 | 44.4 | EN |
| Pests and diseases | [1] Null | 16.6 | 33.3 | EN |
| | Medium | 2 | 0 | EN |
| | High | 81.4 | 66.6 | EN |

[1] Selected as an indicator of sustainability in Figure 2. [2] EN: Environmental dimension.

### 3.2.3. Adaptability

- This attribute was the main bottleneck for both municipalities. Although the smallholder farmers noticed a climate change, they were not aware and had not yet adopted some new strategies. The conservation of soil was a concern for few, but for all of them, water supply was the most important problem. However, they did not have a plan for water management, nor for the diversification of water sources. Moreover, in both municipalities, traditional knowledge was scarcely transmitted to the new generations, and it is being lost as the youths abandon both the farmlands and their villages (Table 5).

**Table 5.** Adaptability indicators for the El Cardonal and Ixmiquilpan municipalities.

| Indicator | Class | Ixmiquilpan | Cardonal | [2] Sustainability Dimension |
|---|---|---|---|---|
| Income diversification | [1] Low | 72.3 | 44.5 | S |
| | Medium | 27.7 | 55.5 | S |
| | High | 0 | 0 | S |
| Access to extension services | [1] High | 16.6 | 22.2 | S |
| | Low | 77.7 | 37.14 | S |
| | Null | 5.5 | 29.62 | S |
| Generational renewal | [1] Youth workers 18–34 years | 5.5 | 3.7 | S |
| | Adult workers 35–59 years | 72.2 | 62.96 | S |
| | Senior workers ≥60 years | 22.2 | 33.34 | S |
| Educational level | Uneducated | 0 | 14.81 | S |
| | Elemental and junior high school | 61.1 | 66.6 | S |
| | | 38.8 | 18.51 | |
| | [1] Senior high school and college | 38.8 | 18.51 | S |
| Impact of global climate change (GCC) | Aware of GCC with impact | 16.6 | 14.81 | EN |
| | [1] No impact | 38.8 | 40.74 | EN |
| | Unknown | 44.4 | 44.4 | EN |
| Management of natural resources | [1] Optimal | 50 | 25.92 | S |
| | Medium | 44.4 | 25.92 | S |
| | Non-conservation | 5.5 | 48.14 | S |

**Table 5.** *Cont.*

| Indicator | Class | Ixmiquilpan | Cardonal | [2] Sustainability Dimension |
|---|---|---|---|---|
| Generational transmission | [1] High | 5.5 | 3.7 | S |
| | Low | 72.2 | 62.96 | S |
| | Null | 22.2 | 33.3 | S |
| Technological adoption | [1] High | 27.7 | 11.1 | S |
| | Low | 61.1 | 81.48 | S |
| | Null | 11.1 | 7.40 | S |

[1] Selected as an indicator of sustainability in Figure 2. [2] EN: Environmental dimension, S: Social dimension.

### 3.2.4. Equity

- Equity focuses mainly on social justice and profit distribution. The Gini coefficient indicated an inequity in the income distribution with values of 0.26 for Ixmiquilpan and 0.44 for El Cardonal. In Ixmiquilpan, this coefficient was the same for the irrigation and rainfed systems, but it was different in El Cardonal; the values were 0.24 for irrigated lands and 0.46 for rainfed lands. Indeed, since the profit equity was higher in El Cardonal for rainfed agriculture, the introduction of irrigation systems could have a negative impact on equity (Table 6).
- Women worked in agriculture in both municipalities; indeed, 27 percent of farmers were women in Ixmiquilpan and 18 percent were women in El Cardonal (Table 6). However, women still faced cultural and legal discrimination, such as a lack of access to land, financing, markets, agricultural training, and education, as well as suitable working conditions and equal treatment; therefore, they were at a disadvantage. The migration of young males gave young women the opportunity to access higher education levels; thus, they are on the way to becoming empowered.

**Table 6.** Equity indicators for the El Cardonal and Ixmiquilpan municipalities.

| Indicator | Class | Ixmiquilpan | Cardonal | [2] Sustainability Dimension |
|---|---|---|---|---|
| Infrastructure | [1] High | 17.1 | 18.51 | E |
| | Medium | 68.5 | 44.4 | E |
| | Null | 14.2 | 37.03 | E |
| Agricultural advisory services | Request | 57.14 | 62.96 | S |
| | [1] Did not request | 40 | 33.3 | S |
| | Null | 2.8 | 3.7 | S |
| Permanent workforce | [1] Family members per ha | 1.43 | 1.95 | S |
| Demand for temporary workforce | [1] Wages per ha | 2.16 | 2.29 | S |
| Distribution of machinery and vehicles | [1] Own (tractor) | 15.6 | 7.4 | E |
| | Borrowed (tractor) | 5.5 | 3.7 | E |
| | [1] Rented (tractor) | 61.1 | 77.7 | E |
| | Null (tractor) | 16.6 | 11.1 | E |
| | Own (vehicle) | 77.7 | 66.6 | E |
| | Borrowed (vehicle) | 0 | 0 | E |
| | Rented (vehicle) | 0 | 0 | E |
| | Null (vehicle) | 22.2 | 33.3 | E |
| Income distribution (Gini coefficient) | [1] Total agricultural activity | 0.2623 | 0.4455 | E |
| | Irrigation system | 0.2296 | 0.2494 | E |
| | Rainfed system | 0.2234 | 0.4681 | E |
| Women's participation | [1] Woman | 27.7 | 18.51 | S |

[1] Selected as an indicator of sustainability in Figure 2. [2] E: Economic dimension, S: Social dimension.

### 3.2.5. Self-Management and Self-Sufficiency

- The autonomy of farmers in controlling their crops and household economy refers to self-management and self-sufficiency. A farmer's decision to use external agricultural supplies depends on the price and availability of seeds, agrochemicals, and other products with local suppliers. Everyone purchased what they could find by themselves, usually at a very high price, because they were not organized. In Ixmiquilpan, more than half of the farmers used chemical fertilizers, and in El Cardonal, only 11.1 percent used them. However, in both municipalities, 44.4 percent of the farmers said that they used pesticides (Table 7). Pesticides in El Cardonal were used for the control of pests in agaves rather than staple crops, while in Ixmiquilpan, pest control through fumigation and use of chemical fertilizers was a common practice in irrigated systems (Table 7).

- Corn is the main food staple in Mexico; Mexicans consume an average of 200 kg per person per year [37]. The corn-fractional food self-security (i.e., the extent to which the farmers can satisfy the corn needs of their families with their own crops) was sufficient for 51% of farmers in El Cardonal and 70% in Ixmiquilpan (Table 7). Thus, drought may have a greater effect on food security in El Cardonal than in Ixmiquilpan.

**Table 7.** Self-management and self-sufficiency for the El Cardonal and Ixmiquilpan municipalities.

| Indicator | Class | Ixmiquilpan | Cardonal | [2] Sustainability Dimension |
|---|---|---|---|---|
| Pesticides | Yes | 44.4 | 44.4 | EN |
| | [1] No | 55.6 | 55.6 | EN |
| Chemical fertilizers | Yes | 61.1 | 11.1 | EN |
| | [1] No | 38.8 | 88.8 | EN |
| Fractional food security level (corn kg/population) | [1] Higher ≥200 | 70 | 51.8 | S |
| | Average 199–100 | 2 | 3.7 | S |
| | Under <100 | 28 | 44.5 | S |
| Agricultural income | [1] High (>USD 1.5 K/year) | 94.4 | 77.7 | E |
| | Medium (USD 1.5 K–USD 770 per year) | 5.5 | 18.5 | E |
| | Low (<USD 770 per year) | 0 | 3.7 | E |
| Dependence on external inputs | High (2 or more supplies) | 44.4 | 7.4 | E |
| | Medium (one supply) | 27.7 | 40.7 | E |
| | [1] Null | 27.7 | 51.8 | E |
| Organization | [1] Intention | 66.6 | 62.96 | S |
| | No intention | 33.4 | 40.74 | S |

[1] Selected as an indicator of sustainability in Figure 2. [2] E: Economic dimension, S: Social dimension, EN: Environmental.

### 3.2.6. Radar Chart

Indicators selected from Tables 3–7 were used to identify the critical points of each attribute in the radar chart (Figure 2). Each indicator was plotted on a relative scale from 0 to 100; the optimal value for sustainability expected for the radar chart was 100. The adaptability was minimal for both municipalities, since all indicators were below 40 percent, excepting income diversification, because farmers the farmers carried out several activities besides agriculture. The stability–resilience–reliability attribute had two indicators that were close to the optimal value; these were corn seed variety and fertilizers for El Cardonal,

and the latter for Ixmiquilpan. This optimal value for the corn seed variety was associated the native corn seeds cultivated in the rainfed systems of El Cardonal. On the other side, the farmers of Ixmiquilpan used commercial seeds in irrigated systems and native varieties in their backyards. Solid livestock wastes were handled as fertilizer in both municipalities. There was an important difference between both municipalities in the stability–resilience–reliability attribute; this difference was due to the use of commercial varieties of crops and fruit trees in Ixmiquilpan and the use of native seeds and native varieties in El Cardonal.

For the next attributes, three indicators of equity were over 50% (permanent workforce, distribution of machinery and vehicles, and temporal workforce) in both municipalities. With respect to the external inputs, El Cardonal had a dependence of less than 50 percent, while it was 73 percent for Ixmiquilpan. In the case of self-management, the most relevant indicator for Ixmiquilpan was the average agricultural income with a value of 92 percent, while for El Cardonal, it was the minimal use of chemical fertilizers. On the other hand, the organization indicator was similar for both municipalities, as well as the cost–benefit ratio, although the total agricultural sales were slightly higher for Ixmiquilpan (Figure 2).

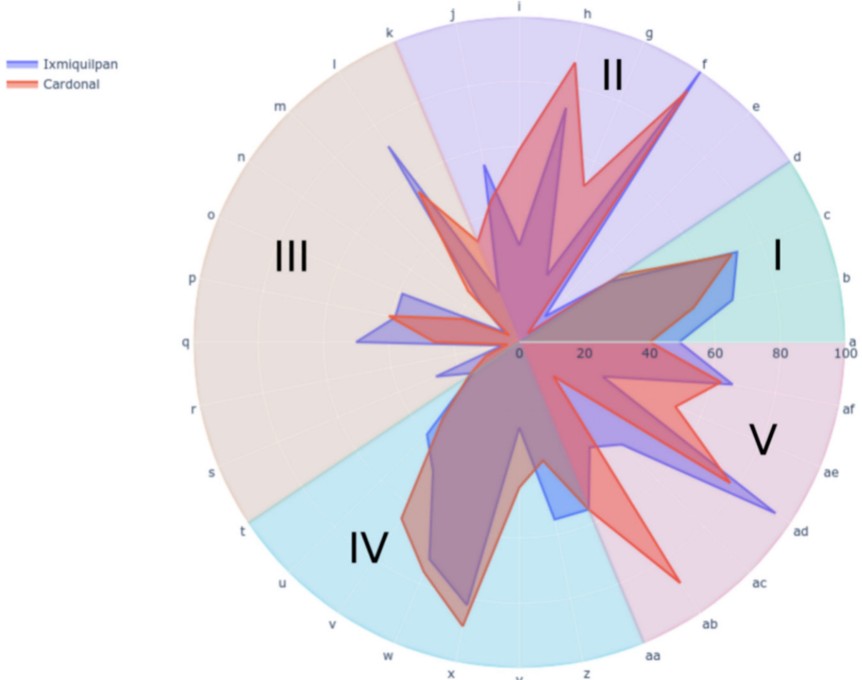

**Figure 2.** Indicators of sustainability for the Ixmiquilpan and El Cardonal municipalities. Each colored sector corresponds to an attribute; starting with the indicator "a" and continuing in an anticlockwise direction, the attributes are: (I) productivity, (II) stability, resilience, and reliability, (III) adaptability, (IV) equity, and (V) self-management and self-sufficiency. The indicators are: (a) total agricultural sales (irrigation), (b) total agricultural sales (rainfed), (c) cost–benefit Ratio (CBR), (d) agricultural supplies, (e) water access, (f) fertilizers, (g) agrobiodiversity, (h) corn seed variety, (i) seed variety of other crops, (j) agroecological practices, (k) pests and diseases, (l) income diversification, (m) access to extension services, (n) generational renewal, (o) educational level, (p) impact of global climate change (GCC), (q) management of natural resources, (r) generational transmission, (s) new technology adoption, (t) infrastructure, (u) agricultural advisory services, (v) permanent workforce, (w) demand for temporary workforce, (x) distribution of machinery and vehicles, (y) income distribution (Gini coefficient), (z) women's participation, (aa) pesticides, (ab) chemical fertilizers, (ac) corn-fractional food security level (corn kg/population), (ad) income from agriculture, (ae) dependence on external inputs, and (af) organization.

Because rainfall is minimal in both municipalities (lower than 150 mm per year), water availability is critical for agriculture and, thus, for stability, resilience, and reliability

(Figure 2). This indicator was higher in El Cardonal than in Ixmiquilpan because the water sources were rainfall and underground water (7% of water availability [39]) in the former and wastewater from Mexico City in the latter.

## 4. Discussion

The evaluation of sustainability through the MESMIS methodology involves different attributes that allow us to design strategies and identify critical points for the implementation of programs for agricultural sustainability [24].

The critical points identified in both municipalities belong to the attributes of adaptability, equity, and self-management and self-sufficiency. All of these attributes indicate the fragility of economic and food security, which is associated with the inequity in income distribution and low corn-fractional food security (Figure 2). Fragility is understood as the vulnerability of small farm holders to changes or uncertainty in the economic, social, and environmental conditions [40]. The external factor of migration has an important impact on the fragility of both economic and food security (Figure 3); indeed, in 2015, the injection of remittances represented from 9.0 to 14.6% of the gross domestic product of Ixmiquilpan and 14.6 to 36.2% in El Cardonal [41]. In both municipalities, the agricultural sales are low, but the total incomes are high; therefore, the cost–benefit ratio is in a medium range. This discrepancy is due to migrant remittances.

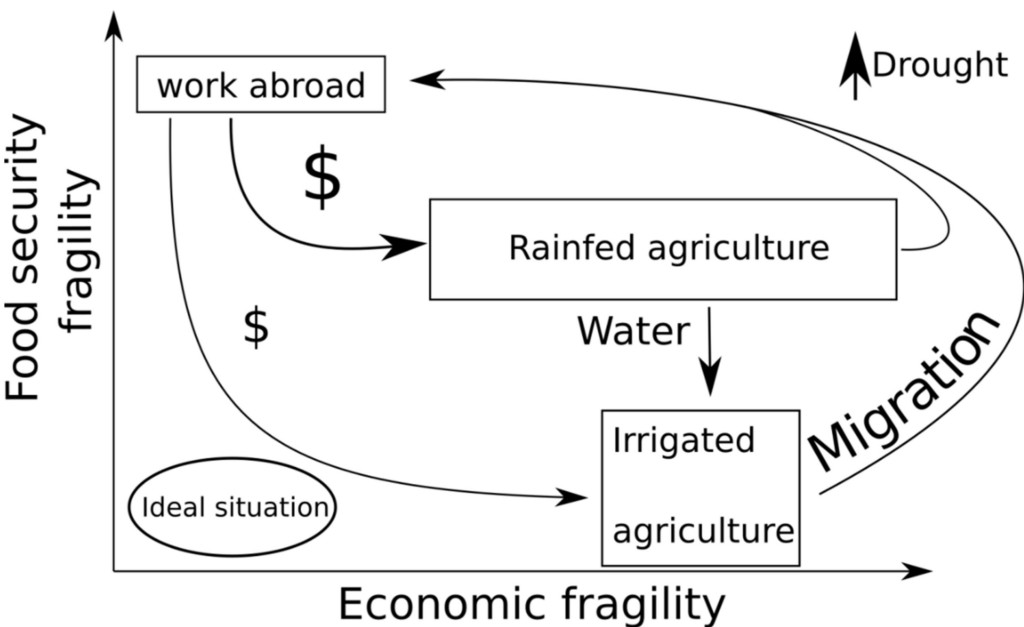

**Figure 3.** Model of interactions between the fragilities of economic and food security.

The small farm holders in Ixmiquilpan depend on external agricultural supplies, and their agricultural infrastructure is scarce, but the remittances are not used for improving this infrastructure, for agriculture innovation, or acquiring supplies; in fact, the remittances are mainly used for purchasing food and for satisfying other necessities, as well as building houses for the family. Thus, remittances lead to an unsustainable economic cycle, thus increasing the fragility of the agricultural system (Figure 3). On the other hand, in El Cardonal, the injection of money does not compensate for the low agricultural yields, and the producers must diversify their activities to be able to satisfy their food necessities; as a result, they neglect their farmlands. Thus, remittances increase the vulnerability of agriculture for a long time. Furthermore, the increase in the fragility of food security is affected by the insufficiency of water, decrease in native corn consumption, and increase in industrialized food products. Previous studies of mesquite trees (*Prosopis laevigata* Humb. et Bonpl. ex Willd) in a community of Ixmiquilpan [42] also pointed to the influence of migration

on the stability and reliability attributes and on the lack of generational transmission of knowledge, which affects the adoption of traditional techniques by new generations.

In fact, remittances may have positive and negative effects in rural Mexican communities. Edward Taylor et al. studied the impact of migrant remittances on the distribution of rural income and rural poverty in Mexico [43]. They found that remittances from migrants abroad slightly increase rural income inequalities, while remittances of internal migrants are equalizers of income, but in high-migration regions both types of remittances have an equalizing effect on income in the long term. Ixmiquilpan is a high-migration municipality, and migration is even higher in El Cardonal; therefore, remittances may have an equalizing effect on income. Hikmet Ersek, President and Chief Executive Officer of Western Union, said that remittances generate crucial positive economic and social effects in developing countries. However, we found that remittances in agricultural systems reduce the likelihood of family members continuing agricultural activities in the long term; they also have a negative effect on cultural identity and on the traditional diets of indigenous peoples. Indeed, remittances decrease the possibility of transferring traditions and cultural practices to younger generations, and they affect the adoption of new technologies and the appropriation of new solutions for improving agricultural yields, water resource management, and sustainability. Thus, remittances from abroad have at least two faces. Similar results were found in Juruvita in the Boyocatá department of Colombia, where farmers had severe difficulties in maintaining agricultural production due to migration and the diversification of their activities in order to satisfy their families' necessities. However, a high percentage of food products came from their own production, and they did not purchase as many products as the farmers from El Cardonal did [24].

On the other hand, the use of untreated wastewater from Mexico City in the agricultural irrigation systems of Ixmiquilpan impacts agriculture in both positive and negative ways. On one side, it has a negative effect on the inhabitants' health and increases their vulnerability to climate change [44]. Indeed, the increase in agricultural yields due to wastewater irrigation is linked to large investments in supplies, unsafe food products, diseases, and malnutrition; finally, it affects the farmers' income [45]. On the other side, wastewater has, in the short term, a positive effect on agricultural yields. However, in the environmental context, there are other options for using and regenerating dry lands—for example, Luján Soto et al. recommended a combination of techniques [46], such as the reforestation of local crops, including agave or cactus, and technical services for the control of plagues and phytopathogens. In this context, the adoption of climate-smart agriculture has impacts on carbon sequestration and biodiversity conservation; in fact, it could generate additional incomes for the smallholders through ecosystem services [47].

In summary, (1) adaptability is a critical factor in both municipalities and it does not depend on economic factors; rather, the main problem is the migration of youths. (2) Access to water and economic resources and the management of environmental resources are an necessary in order to increase the yield of agricultural crops and equity. (3) The strengthening of resilience requires organizations of small producers and the combination of indigenous and modern technologies for territorial development. (4) The social adaptation of both communities is at a critical level because of the generational break between farmers and their offspring. (5) Productivity depends mostly on agricultural supplies, thus affecting farmers' CBR, which is similar for rainfed and irrigation agriculture. (6) Remittances have two faces—a positive effect on income and poverty reduction and a negative one on agriculture, traditional diets, and cultural identity (Figures 2 and 3).

## 5. Conclusions

From the integrative analysis and comparison of the two agricultural systems (rainfall and irrigation) in the same dry-land region, our new finding is the important difference between income distribution and the management of the water resources. Notably, the main factor that affects income distribution is migration, and remittances influence the fragility of both types of agricultural systems, food security, and cultural identity.

The vulnerability of the agricultural systems to crop pests and diseases must be addressed, as it reduces productivity and the associated profits, forcing farmers to seek other economic activities. Furthermore, technical services are also needed in order to apply other agroecological practices for solving problems, such as optimal management of water resources and pest and disease control. The lack of technical services, together with the economic problems, contributes to the breaking of the generation renewal and forces the youth to migrate to find a better life.

Some recommendations for both municipalities to reduce the fragility of their food security are the implementation of integral systems for water management, including the capture of rainfall, underground water, and treated wastewater, as well as the implementation of efficient irrigation systems for decreasing water and energy consumption. These innovations should integrate traditional knowledge and modern science and technology by including producers, youths, men, and women. The organization of small farm holders is crucial for the development of long-term and sustainable projects.

Other recommendations for long-term sustainability are (1) the promotion of polycultures in the region by using native staple crops to increase food security, leading to a robust economic income, (2) the promotion of sustainable rural enterprises—to increase the values of agricultural products, organizations of producers and their families must own these companies—and (3) appropriation by producers of the supply chain and distribution channels.

These recommendations are aimed at increasing the economic sustainability of agriculture and its resilience to climate change. More robust and profitable crops would be more attractive to young farmers, which would stimulate cross-generational continuity. In that context, migration would turn from a necessity into an option for youths. In addition, more productive agricultural systems would encourage farmers and their relatives to invest in agriculture. Thus, by strengthening the agricultural system, the fragility of the socioeconomic system of the region would be improved overall, with less migration and more regional benefits. Finally, a strengthened agriculture could bring robustness for the region in the case of macroeconomic disturbances, as it would ensure food security in terms of self-nourishment.

This evaluation of sustainability is only a picture of the current agricultural situation; therefore, it is essential to measure the sustainability indicators over a period of time in order to evaluate and adjust the strategies and recommendations for governmental policies. Therefore, we plan to improve the statistical sampling with a temporal comparison of data over a short period as future research in order to improve our recommendations and strengthen the agricultural sustainability in this region and other dry-land regions.

**Author Contributions:** Conceptualization, Y.C. and M.D.l.T.; methodology, Y.C.; validation, Y.C., D.L., and M.D.l.T.; formal analysis, Y.C.; investigation, Y.C.; resources, D.L.; data curation, M.D.l.T. and Y.C.; writing—original draft preparation, Y.C.; writing—review and editing, M.D.l.T., D.L., and Y.C.; visualization, Y.C.; funding acquisition, D.L. All authors have read and agreed to the published version of the manuscript.

**Funding:** This research was funded by PRODETER, grant number SDA/SDR/CITT/014/2019, PRODETER, grant number SDA/SDR/CITT/015/2019. This work was supported by a Newton Fund Institutional Links grant, ID 628849219, under the Newton-Mexico partnership. The grant is funded by the UK Department for Business, Energy and Industrial Strategy and Consejo de Ciencia, Tecnología e Innovación de Hidalgo (CITNOVA) and Catedras-CONACYT, project number 485.

**Institutional Review Board Statement:** Not applicable.

**Informed Consent Statement:** Informed consent was obtained from all subjects involved in the study.

**Data Availability Statement:** The data presented in this study are available on request from the corresponding author.

**Acknowledgments:** The authors would like to thank E. Aguirre von Wobeser and Pablo Wong for their valuable recommendations. The inhabitants of Ixmiquilpan and El Cardonal municipalities are thanked for their willingness to participate. Thanks are given to the team of the Regional Unit Hidalgo of Centro de Investigacion en Alimentacion y Desarrollo A. C. for the collection of data.

**Conflicts of Interest:** The authors declare no conflict of interest.

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
