# Peer review of "Sustainability of the Agricultural Systems of Indigenous People in Hidalgo, Mexico"

_sustainability, doi:10.3390/su13148075_

Round 1

Reviewer 1 Report

In the manuscript the authors presented an important point, although the manuscript has some drawbacks.

Main remarks:

1. Intruduction - what is the research gap?

2. Literature review - after the "introduction", there should be a part of "literature review". As a result, the number of scientific publications used in the manuscript will increase, which in my opinion is only 37.

3. Materials and Methods - in my opinion, there are no in-depth taxonomic and statistical methods.

4. Conclusions - in your conclusions, please also answer the following questions:

• what are the directions for the future?
• what are the research gaps?
• what is new to this manuscript?

Reviewer 2 Report

Overall, the paper looks good and the area of study, i.e. "Sustainability of agricultural systems of indigenous people" is very timely. However, analysis using descriptive statistics only is difficult to justify. 

There is ambiguity in definitions of parameters (indicators), and their classes. For example, for access to extension services, the parameter was defined as high when peasants received technical assistance, low when they did not receive it but had requested it, and null when they did not receive assistance and they did not know with certainty if they needed it. It is unclear whether peasants who were assigned "high" scores were those who received technical assistance after they requested it, or technical assistance was provided to them without them requesting it. Further, reviewers like me and even readers would also want to know what did it mean by "technical assistance" because technical assistance mean different things to different people.

Similarly, for technological adoption the percentage of producers who had access to cell phone, the Internet, etc. and were assigned high, low, and null. Here the question comes, for those receiving "high" score, did they have access to phone or the Internet or both or other IT tool or even those with access to only one tool received a "high" score. Similar questions are  for "low" and "null."

The definitions of parameters and their indicators should be mutually exclusive so that all respondents understand and/or interpret these parameters and indicators similarly. This is critical to avoid biases/errors in data.

If I am not wrong, only 121 peasants (15 or 20%) were selected through non-probability sampling and were interviewed. It is unclear how many respondents/samples were from each municipalities. It is possible that those who did not volunteered to participate in the study had different opinions or characteristics than those who participated. Therefore, please justify your sampling design.

Round 2

Reviewer 1 Report

Accept in present form. Good luck!

Reviewer 2 Report

The paper looks much improved than the previous version. However, there are a few places requiring grammatical corrections.
